

# Characterization of trade-offs between immunity and reproduction in the coral species *Astrangia poculata*

Natalie Villafranca, Isabella Changsut, Sofia Diaz de Villegas, Haley Womack and Lauren E. Fuess

Department of Biology, Texas State University, San Marcos, TX, United States

## ABSTRACT

**Background:** Living organisms face ubiquitous pathogenic threats and have consequently evolved immune systems to protect against potential invaders. However, many components of the immune system are physiologically costly to maintain and engage, often drawing resources away from other organismal processes such as growth and reproduction. Evidence from a diversity of systems has demonstrated that organisms use complex resource allocation mechanisms to manage competing needs and optimize fitness. However, understanding of resource allocation patterns is limited across taxa. Cnidarians, which include ecologically important organisms like hard corals, have been historically understudied in the context of resource allocations. Improving understanding of resource allocation-associated trade-offs in cnidarians is critical for understanding future ecological dynamics in the face of rapid environmental change.

**Methods:** Here, we characterize trade-offs between constitutive immunity and reproduction in the facultatively symbiotic coral *Astrangia poculata*. Male colonies underwent ex situ spawning and sperm density was quantified. We then examined the effects of variable symbiont density and energetic budget on physiological traits, including immune activity and reproductive investment. Furthermore, we tested for potential trade-offs between immune activity and reproductive investment.

**Results:** We found limited associations between energetic budget and immune metrics; melanin production was significantly positively associated with carbohydrate concentration. However, we failed to document any associations between immunity and reproductive output which would be indicative of trade-offs, possibly due to experimental limitations. Our results provide a preliminary framework for future studies investigating immune trade-offs in cnidarians.

# INTRODUCTION

The ability to successfully defend against a variety of pathogenic threats is an essential component of organismal fitness (*Lochmiller & Deerenberg, 2000*). However, most immune defenses which prevent or mitigate damage from pathogens are energetically costly (*Lochmiller & Deerenberg, 2000*). This is particularly important as organisms

Corresponding author
Lauren E. Fuess, lfuess@txstate.edu

operate using a fixed budget of energetic resources that must be allocated to competing demands (*i.e.*, growth, reproduction, defense, *etc*; *Stearns, 1989*). Consequently, allocation of resources to one demand often comes at the cost of others, creating trade-offs (*Melbinger & Vergassola, 2015*). Optimal organismal fitness therefore requires careful allocation of resources across these demands to minimize negative effects of trade-offs and maximize reproductive output. Notably, the high costs of maintaining immune systems and mounting immune responses often result in trade-offs with other traits (*Lochmiller & Deerenberg, 2000*; *Martin, Weil & Nelson, 2008*; *Rauw, 2012*; *Rigby & Jokela, 2000*; *van der Most et al., 2010*). Among the most well documented immune-related trade-offs are those between reproduction and immunity, with organisms most often able to exhibit high immune capacity or high reproductive ability, but not both (*Adamo, Jensen & Younger, 2001*; *Brokordt et al., 2019*; *Gwynn et al., 2005*; *Hosken, 2001*; *Schwenke, Lazzaro & Wolfner, 2016*). However, despite frequent empirical observation of immune-reproductive trade-offs and robust proposed theory, many questions still remain regarding the generalizability of resource allocation theory and immune-related trade-offs. In some systems, consistent immune-reproductive trade-offs have not been observed (*Kelley et al., 2021*; *McNamara et al., 2013*; *Miyashita et al., 2019*; *Syed et al., 2020*; *Xu, Yang & Wang, 2012*). Furthermore, it remains unclear how dynamic changes in resources/energetic budget might impact these trade-offs. It is theorized that since total energy and relative allocation are co-dependent, variation in energy budget may have an impact on trade-offs such as those between immunity and reproduction, however limited empirical evidence exists to support or refute this theory (*Cotter et al., 2010*; *Descamps et al., 2016*; *Simmons, 2012*; *Stahlschmidt et al., 2013*). Combined, these gaps in existing theory make it difficult to predict how resources are allocated among traits, requiring further study to improve understanding of resource allocation and immune trade-offs in complex contexts. This is particularly important in light of the rising prevalence of epizootic outbreaks affecting vulnerable species across the globe (*Croquer & Weil, 2009*; *Glenn & Pugh, 2006*; *Kilburn et al., 2010*).

Scleractinian or reef building corals are arguably one of the taxa most impacted by increases in epizootic outbreaks (*Bruno et al., 2007*; *Precht et al., 2016*; *Ruiz-Moreno et al., 2012*). Disease outbreaks have been one of the largest drivers of coral declines in recent decades, affecting almost all major reef building species (*Alvarez-Filip et al., 2022*; *Bourne et al., 2009*; *Cróquer, Weil & Rogers, 2021*; *Moriarty et al., 2020*; *Sharma & Ravindran, 2020*; *Sutherland, Porter & Torres, 2004*). Still, despite the severity of these outbreaks, many gaps remain in our understanding of associated disease processes. For example, significant inter- and intra-specific variation has been observed in coral disease resilience (*Alvarez-Filip et al., 2022*; *Miller et al., 2019*; *Palmer et al., 2011a*). While some studies have pointed to the importance of divergence in certain cellular processes in driving this variation (*Beavers et al., 2023*; *Fuess et al., 2017*; *MacKnight et al., 2022*; *Pinzon et al., 2014a*; *Traylor-Knowles et al., 2021*), little is known regarding how broader ecological processes, including resource allocation, might contribute to variation in disease resistance. To date, most studies of cnidarian resource allocation have focused on stress associated trade-offs. The negative effects of thermal anomalies on coral reproductive output have been well

documented (*Michalek-Wagner & Willis, 2001*; *Nielsen et al., 2020*; *Paxton et al., 2016*). In contrast, only a handful of studies have documented reproductive trade-offs involving immunological processes, all of which had focused on the negative impacts of disease outbreaks on coral reproduction (*Fuess et al., 2018*; *Weil, Croquer & Urreiztieta, 2009*). No studies have directly linked reproductive investment and constitutive immunity in cnidarians, in part due to the historic lack of tractable model cnidarian systems.

Recent advances in the development of cnidarian model and study systems have greatly expanded the scope of cnidarian research. One such emergent model species is *Astrangia poculata* (common name: northern star coral), a non-reef building stony coral that can be found along the Atlantic Coast of the United States ranging from Cape Cod, MA to the Texas Gulf Coast (*Dimond et al., 2012*). Importantly, unlike tropical corals which are dependent on symbiotic relationships for survival, *A. poculata* associates facultatively with a single species of symbiont, *Breviolum psygmophilum* (*Lajeunesse, Parkinson & Reimer, 2012*) with colonies ranging from high symbiont density ("brown") to low symbiont density ("white"; *Sharp et al., 2017*). Classification of symbiotic states is based on color, approximate chlorophyll concentration, and symbiont density ("brown" >$10^6$ cells cm$^{-2}$; "white" $10^4$–$10^6$ cells cm$^{-2}$; *Sharp et al., 2017*). A key benefit of the symbiotic relationship between corals and their symbionts is the exchange of organic nutrients to the host (*Kirk & Weis, 2016*). Consequently, facultative symbiosis may serve as a natural system for exploring the effects of variable resource budget on immune-associated trade-offs in corals, though the exact effects of variation in symbiont density on host energetic budget are poorly studied (*Szmant-Froelich & Pilson, 1980*). In contrast, recent studies have highlighted significant impacts of variation in *A. poculata* symbiont density on host immunity (*Changsut et al., 2022*; *Harman et al., 2022*), suggesting that variation in host-symbiont dynamics have significant consequences for broader organismal physiology.

Here we used the tractable *A. poculata* study system to investigate cnidarian resource allocation and potential trade-offs between reproduction and immunity (*i.e.*, negative associations between reproduction and immunity resulting from resource limitation). We first assessed general effects of variability in symbiont density on host physiology (reproductive output, energetic budget, and immune activity). Next, we assessed the effects of energetic budget on both reproductive output and immune activity generally, independent of symbiont density. Finally, we tested for reproductive-immune trade-offs. Our study provides a preliminary framework for future studies considering the inter-connected roles of symbiosis, resource allocation, and immunity in cnidarians.

## MATERIALS AND METHODS

Portions of this manuscript were previously published as a preprint (*Villafranca et al., 2023*).

### Sample collection & experimental design

Entire colonies of "white" (low symbiont density; $n$ = 13) and "brown" (high symbiont densities; $n$ = 7) *A. poculata* ranging in size from ~5–61 cm (17–86 polyps) were collected

from Fort Wetherill in Jamestown, Rhode Island (41°28′40″N, 71°21′34″W) in August 2021, at a depth of 10–15 m. Corals were collected under RI Department of Environmental Management Permit #825 issued to Koty Sharp/Roger Williams University. Previous studies have indicated *A. poculata* reaches peak gametogenesis between August and September each year (*Szmant-Froelich, Yevich & Pilson, 1980*). Samples were returned to Roger Williams University (Bristol, Rhode Island), and held overnight prior to experimental spawning. To trigger spawning, individuals were moved from the recirculating tanks at ambient temperatures (~19 °C) to individual containers with 100 mL of filtered seawater heated to 27 °C. Colonies were then closely monitored for signs of spawning. Colony sex was confirmed based on gametes released at time of spawning (*A. poculata* is gonochoristic; *Szmant-Froelich, Yevich & Pilson, 1980*). Twenty male colonies were identified and used for the study, 13 white and seven brown. Individuals were allowed to spawn for 30 min, at which point gamete release had nearly or completely ceased. Following this period adult colonies were removed and flash frozen in liquid nitrogen. The water in the container was then thoroughly mixed and a 1 mL sample was taken from each colony for sperm density estimation. Sperm density was estimated by counting in triplicate on a haemocytometer; counts were then normalized to number of polyps per colony prior to statistical analyses. Flash frozen adult corals were later shipped to Texas State University for downstream sample processing and analysis.

## Symbiont density quantification

Flash frozen adult colonies were airbrushed using a Paasche air brusher (VL0221) with 100 mM Tris + 0.5 mM Dithiothreitol (DTT; pH 7.8) to remove tissue from the skeleton and extract symbionts and host-enriched proteins (*Fuess et al., 2016*). First, in order to estimate symbiont density, an area of 2.14 cm$^2$ (1-2 polyps) was marked on a flat surface of the colony. Tissue was airbrushed from the area until no tissue remained. The resulting tissue slurry was placed in a 2.0 mL microcentrifuge tube, and homogenized for 10 seconds using a handheld homogenizer (Fisherbrand 150). Following homogenization, samples were centrifuged (2000 RPM for 3 min) and then washed with 500 μl of DI H$_2$O. This process was repeated twice; final samples were preserved in 500 μl of Deionized H$_2$O + 0.01% Sodium Dodecyl Sulfate (SDS) and stored at 4 °C. Symbiont density was later determined *via* microscopy (Nikon Eclipse E600) by counting in triplicate on a hemocytometer (*Changsut et al., 2022*; *Mieog et al., 2009*).

## Protein extraction and immune metric analyses

A cell free tissue extract was generated from the remaining portion of the *A. poculata* colony following established protocols (*Changsut et al., 2022*). Host immunity was then characterized using a suite of established biochemical immune assays designed to measure multiple components of host coral immunity. Specifically we measured antioxidant activity (catalase and peroxidase), components of the melanin synthesis cascade (total phenoloxidase activity and melanin concentration), and antibacterial activity following

established protocols optimized for *A. poculata* (*Changsut et al., 2022*). The assays capture three main components of the coral immune system. Antioxidant activity is essential in both stress and immune response, mitigating the impacts of toxic reactive oxygen species produced by pathogens, as a byproduct of other immune pathways, or as a result of stress (*Palmer, Roth & Gates, 2009*; *Tarrant et al., 2014*). The melanin synthesis cascade is a multi-functional pathway with important roles in immune defense. Melanin can be used to encapsulate pathogens, and is directly cytotoxic to pathogens (*Palmer et al., 2011b*). The production of melanin is dependent on phenoloxidase enzymes which are secreted in an inactive form (*Cerenius et al., 2010*; *Palmer, Mydlarz & Willis, 2008*); total phenoloxidase assays captures activity of both active and inactive phenoloxidases. Finally, we measured the ability of cell free protein extracts to inhibit growth of the known coral pathogen, *Vibrio coralliilyticus* (Strain RE22Sm provided by D. Nelson at University of Rhode Island; *Ushijima et al., 2020*, *2014*) as a metric of antibacterial activity. All immune assays were run in triplicate on 96 well plates using a Cytation 1 cell imaging multi-mode reader and Gen 5 software (BioTek Instruments, Winooski, VT, USA). A Red660 (G Biosciences, St. Louis, Missouri) assay was used to determine protein concentration to standardize immune activity metrics (*Mydlarz & Palmer, 2011*). Full processing and assay details can be found in Supplementary File 1.

## Energetic assays (lipid and carbohydrate concentration)

Energetic budget, specifically lipid and carbohydrate concentration, were estimated using a portion of the generated cell free tissue extract. Measurement of lipid and carbohydrate concentration is a standard approach for estimating coral energetic budget (*Rodrigues & Grottoli, 2007*) A standard coral protocol for quantification of lipids within coral tissue slurries was used to estimate lipid content (*Bove & Baumann, 2021*). Similarly, we estimated total carbohydrate concentration following previously established protocols (*Dubois et al., 1956*; *Masuko et al., 2005*). Both lipid and carbohydrate concentration were standardized by dry tissue weight. Full methodological details can be found in Supplementary File 1.

## Statistical analyses

All statistical analyses were conducted in R (version 4.2.1; *R Core Team, 2021*), with the RStudio integrated development environment (*R Core Team, 2020*). Prior to statistical analyses, outliers were removed based on results of a Rosner Test (EnvStats, v 2.7.0; *Millard, 2013*) One outlier was identified and removed each for peroxidase, melanin, and lipid concentration. Data was checked for normality using a Shapiro Test (base R), and total phenoloxidase activity and sperm density were normalized using a log transformation. We then conducted statistical analyses taking the following approach: first we tested specific hypotheses regarding variation in symbiont density across colony types, effects of variation in symbiont density, and drivers of variation in sperm density. Then we considered the factors contributing to variation in immune function using both multivariate and univariate approaches. Full details of our statistical approaches are below.

## Hypothesis testing

First, differences in symbiont density between white and brown colonies were assessed using a non-parametric t-test (Wilcoxon test; base R) as symbiont density data distribution could not be normalized. Next, the impact of both symbiotic state (categorical) and symbiont density (continuous) on lipid and carbohydrate concentration was evaluated using linear models (base R). Finally, the impact of symbiont density and energetic budget (lipid and carbohydrate concentration) on sperm density was assessed. Using the R package MuMIn (v. 1.47.5; *Barton, 2023*) we dredged all possible linear models explaining this relationship, performed model testing, and then applied model averaging for the best fit models ($\Delta$ AIC < 2).

## Multivariate statistics

Next, multivariate statistics were used to investigate the roles of symbiont state/density, energetic budget (lipids and carbohydrates), and sperm density on immune parameters. First, a MANOVA was run to test for the impacts of symbiont state on immune parameters (base R). Then a principal component analysis was used to further compare immune profiles between white and brown colonies (base R). Prior to PCA analysis all numeric variables were centered and scaled. Next, to determine the effects of our continuous predictors (symbiont density, energetic budget, and sperm density) on immune profiles (response variables) centered and scaled values were used to conduct an RDA analysis with the R package vegan (v. 2.6-4; *Oksanen et al., 2023*). Forward model selection was used to determine significant predictors in our analysis. Finally, a correlational analysis was conducted to test for associations between all continuous variables (predictor and response) using base R (Pearson correlation).

## Univariate statistics

After completing multivariate statistics, the impacts of our predictor variables (symbiont state/density, energetic budget, sperm density) on each of our immune parameters individually were assessed. For each immune parameter the R package MuMIn (v. 1.47.5; *Barton, 2023*) was used to dredge and test all possible linear models. Model averaging was used to summarize all best fit models ($\Delta$ AIC < 2).

# RESULTS

## Hypothesis testing

Symbiont density was significantly higher in colonies classified as "brown" than those classified as "white" (Wilcoxon test, W=91, $p < 0.001$; Fig. S1). However, there was no significant association between symbiont density and carbohydrate (linear model, coefficient estimate = $-3.18e^{-9}$, std err = $7.25e^{-9}$, t val. = $-0.439$, $p = 0.666$) or lipid concentration (linear model, coefficient estimate = $-2.65e^{-10}$, std err = $6.50e^{-9}$, t val. = $-0.041$, $p = 0.968$), nor was there a difference in either carbohydrates (linear model, coefficient estimate = 0.00133, std err = 0.00178, t val = 0.749, $p = 0.463$) or lipids (linear model, coefficient estimate = 0.000383, std err = 0.00131, t val = 0.293, $p = 0.773$) between white and brown colonies. Finally, the only significant predictor of sperm density was

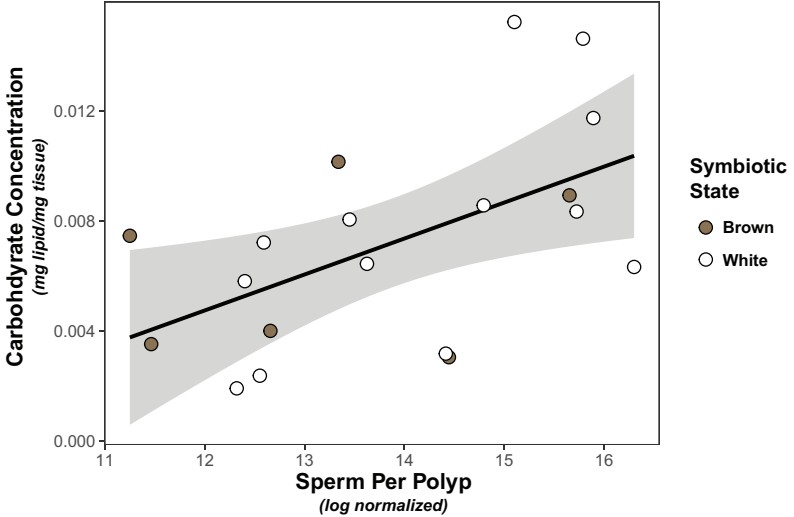

**Figure 1 Linear regression modeling of the relationship between average sperm production per polyp and carbohydrate concentration.** Points are colored based on original sample symbiotic state classification (white or brown). Trendline is representative of the linear model of the relationship of the two variables, with 95% confidence intervals shaded.

**Table 1 Linear model results for sperm density.**

| Predictors | Estimates | SE | t | p value |
|---|---|---|---|---|
| (Intercept) | 12.3 | 0.687 | 17.9 | **<0.001 *** |
| Carbohydrates | 224 | 84.5 | 2.65 | **0.0167*** |

**Note:**
Best-fit linear models for sperm density when including symbiont density, carbohydrate concentration, lipid concentration as predictors. All possible models were compared and model averaging was used where appropriate (AIC delta < 2). Bold font indicates significant p-values; asterisks represent significance: *$p < 0.05$, **$p < 0.01$, ***$p < 0.001$.

carbohydrate concentration, which was positively associated with normalized sperm density (Fig. 1, Tables 1, S1).

## Multivariate statistics

Immunity did not vary significantly as a result of symbiont state (MANOVA, Pillai's Trace = 0.482, F (5/11) = 2.047, $p$ = 0.1497). Principle component analysis was in congruence with this finding (Fig. 2); no clear spatial separation between white and brown colonies was observed. Furthermore, forward model selection for RDA analysis failed to identify any significant predictors. Finally, we identified several significant correlations among our continuous variables (Fig. 3). Carbohydrate concentration was significantly positively correlated to both antibacterial activity (Pearson correlation, $r$ = 0.59, $p$ = 0.0158) and lipid concentration (Pearson correlation, $r$ = 0.51, $p$ = 0.0459). Symbiont density was significantly correlated to melanin concentration (Pearson correlation, $r$ = 0.54, $p$ = 0.0317). Sperm density was significantly positively correlated to total phenoloxidase activity (Pearson correlation, $r$ = 0.5, $p$ = 0.0494).

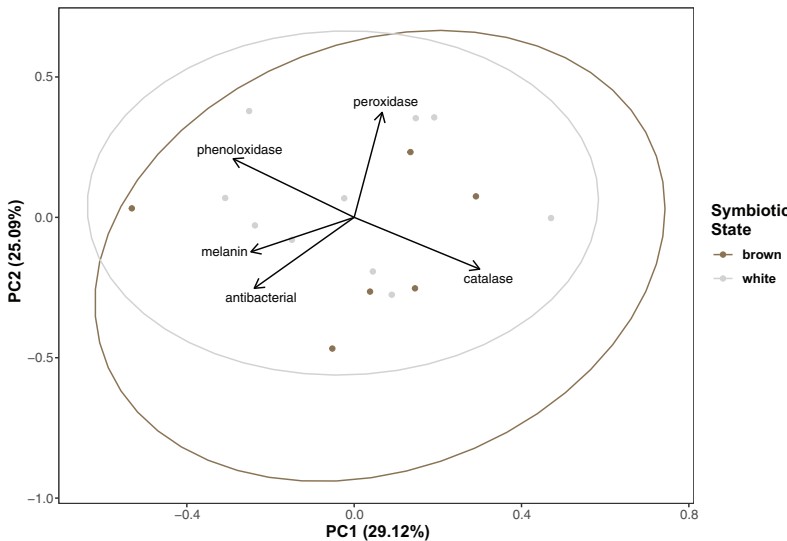

**Figure 2 Principal component analysis showing special orientation of average immune activity of each coral.** Points are colored and grouped by symbiotic state classification. Arrows depict principal component loadings of each measured immune metric.

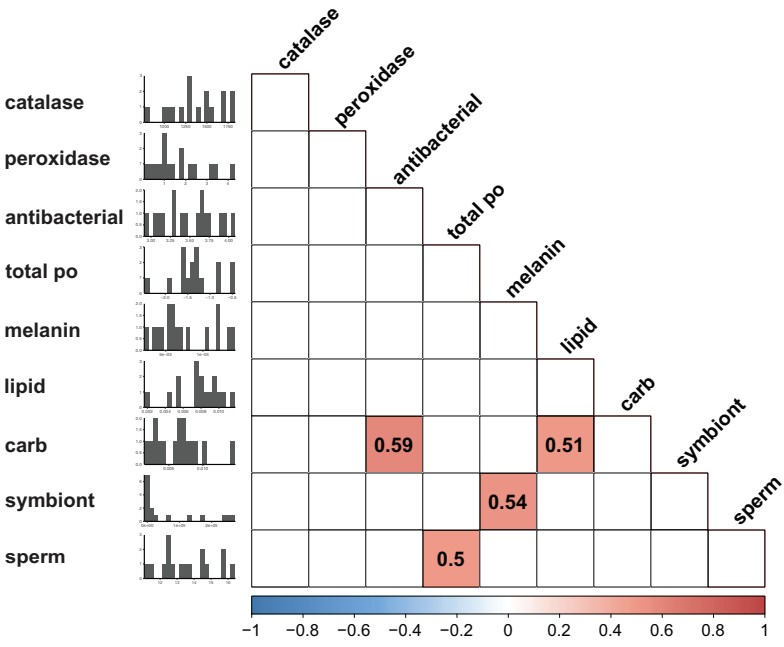

**Figure 3 Correlation plot displaying results of all possible pairwise comparisons of all continuous variables measured (Pearson correlation).** Only significant ($p < 0.05$) associations are displayed. Histograms along the y axis display distribution of each continuous variable (split into approximately 20 bins).

## Univariate statistics

Based on the limited observed significant multivariate associations, we also considered the impacts of each of our predictors (symbiont density/state, energetic budget, sperm density) on activity of our immune metrics using general linear models. Only melanin was

**Table 2  Linear model results for individual immune parameters.**

**Catalase**

| Predictors | Estimates | SE | SEadj | z | p value |
| --- | --- | --- | --- | --- | --- |
| (Intercept) | 1,386 | 134.5 | 141.5 | 9.80 | **<0.001** *** |
| Sperm | $-1.10\text{E}^{-5}$ | $2.02\text{E}^{-5}$ | $2.10\text{E}^{-5}$ | 0.529 | 0.597 |
| Carbohydrates | 9,829 | 18,200 | 18,820 | 0.522 | 0.602 |

**Peroxidase**

| Predictors | Estimates | SE | SEadj | z | p value |
| --- | --- | --- | --- | --- | --- |
| (Intercept) | 1.875 | 0.315 | 0.334 | 5.62 | **<0.001** *** |
| Symbiont | $-2.74\text{E}^{-6}$ | $2.98\text{E}^{-6}$ | $3.08\text{E}^{-6}$ | 0.889 | 0.374 |

**Antibacterial activity**

| Predictors | Estimates | SE | SEadj | z | p value |
| --- | --- | --- | --- | --- | --- |
| (Intercept) | 3.35 | 0.187 | 0.196 | 17.2 | **<0.001** *** |
| Carbohydrates | 13.5 | 20.3 | 20.9 | 0.647 | 0.518 |
| Lipids | 5.32 | 15.0 | 15.5 | 0.343 | 0.732 |

**Total phenoloxidase**

| Predictors | Estimates | SE | SEadj | z | p value |
| --- | --- | --- | --- | --- | --- |
| (Intercept) | $-1.39$ | 0.144 | 0.153 | 9.12 | **<0.001** *** |
| Sperm | $4.06\text{E}^{-8}$ | $3.50\text{E}^{-8}$ | $3.77\text{E}^{-8}$ | 1.08 | 0.282 |

**Melanin**

| Predictors | Estimates | SE | t | p value |
| --- | --- | --- | --- | --- |
| (Intercept) | $2.61\text{E}^{-5}$ | $1.61\text{E}^{-5}$ | 1.621 | 0.126 |
| Carbohydrates | $5.44\text{E}^{-3}$ | $1.82\text{E}^{-3}$ | 2.984 | **0.00927** ** |
| Symbiont | $1.97\text{E}^{-10}$ | $7.37\text{E}^{-11}$ | 2.67 | **0.0176** * |

Note:

Best-fit linear models for each immune parameter when including symbiont density, carbohydrate concentration, lipid concentration, and sperm produced per polyp as predictors. All possible models were compared and model averaging was used where appropriate ($\Delta$ AIC < 2). Bold font indicates significant p-values; asterisks represent significance: *$p < 0.05$, **$p < 0.01$, ***$p < 0.001$.

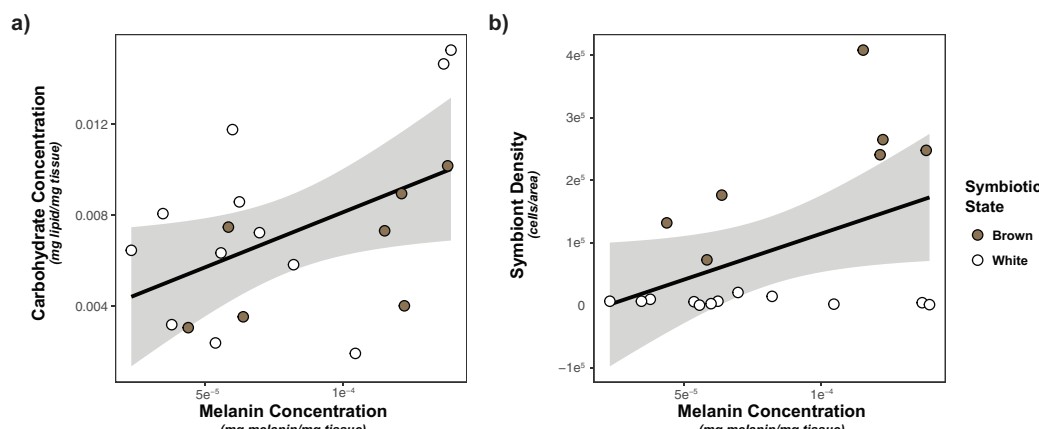

**Figure 4  Linear regression modeling of the relationship between carbohydrate concentration and symbiont density and melanin concentration.** (A) Relationship between carbohydrate concentration and melanin concentration; (B) relationship between symbiont density and melanin concentration. Points are colored based on original sample symbiotic state classification (white or brown). Trendline is representative of the linear model of the relationship of the two variables, with 95% confidence intervals shaded.

significantly impacted by any of predictors (Tables 2, S2). Melanin concentration was significantly positively associated with both carbohydrate concentration ($p = 0.00927$) and symbiont density ($p = 0.0176$; Fig. 4).

## DISCUSSION

Coral reefs globally are undergoing rapid, unprecedented declines due to anthropogenic climate change and associated disease outbreaks (*Hoegh-Guldberg et al., 2007*; *Johnston, Clark & Bruno, 2022*; *Precht et al., 2016*). Corals exhibit both inter- and intra-specific variation in susceptibility to disease and infection, but the ecological factors that drive susceptibility are poorly understood (*Fuess et al., 2017*; *Mydlarz et al., 2016*). Few studies have investigated trade-offs between key life-history traits and immunity in cnidarians (*Alvarez-Filip et al., 2022*; *Pinzon et al., 2014b*; *Schlecker et al., 2022*; *Weil, Croquer & Urreiztieta, 2009*). Here, we used the facultatively symbiotic coral *Astrangia poculata* to investigate potential reproductive-immune trade-offs in scleractinian corals, and the impacts of variation in symbiont density and energy budgets on these trade-offs.

Our study found limited links between variation in symbiont density and our measured physiological traits (lipid/carbohydrate concentration, immune activity, reproductive output). Most notably, we observed no significant difference in total lipid and carbohydrate content as a result of variable symbiont density in adult coral colonies. While this is in agreement with a previous study of *A. poculata* (*Szmant-Froelich & Pilson, 1980*), it is in opposition to conventional thinking regarding symbiosis, which would suggest that increased symbiont density results in increased energetic budget (*Changsut et al., 2022*; *Harman et al., 2022*; *Hughes et al., 2010*; *Pupier et al., 2019*). We propose two alternative hypotheses to explain these patterns: first, it is possible corals with lower symbiont density compensate *via* increased heterotrophic feeding. Unlike tropical corals, *Astrangia* is commonly found in low light, nutritionally rich waters. Consequently, these corals may obtain a significant portion of their nutrition from heterotrophy, especially colonies with low symbiont densities (*Szmant-Froelich, 1981*; *Wuitchik et al., 2021*). Heterotrophic feeding has previously been demonstrated to have a significant role in the efficiency of physiological traits in *Astrangia* and other facultatively symbiotic corals (*Aichelman et al., 2016*; *Dimond & Carrington, 2007*; *Ferrier-Pagès et al., 1998*; *Miller, 1995*). Thus, it is possible that our colonies with lower symbiont densities were compensating *via* heterotrophic feeding and actually not nutrient limited. The absence of nutrient limitation due to heterotrophic feeding may also explain the lack of trade-offs observed (see below). Alternatively, total lipid and carbohydrate assays may not fully captured host energetic budget, as coral energetic budgets are exceptionally complex (*Lesser, 2012*). Other approaches which account for carbon flux, utilize stoichiometric approaches, or assess diverse metabolites in both the presence and absence of heterotrophy, may provide further insight regarding the relationship between symbiont state and energetic budget.

Similar to energetic budget, associations between symbiont density and immunity were limited. Our multivariate analyses failed to detect any differences in immunity between white and brown colonies, nor as a result of symbiont density. However, correlational analysis and univariate linear modeling detected a positive association between melanin

concentration and symbiont density (but not symbiont state). Previous *A. poculata* studies have noted a significant positive association between melanin synthesis and symbiotic state/symbiont density (*Changsut et al., 2022*; *Harman et al., 2022*). This broad consensus regarding the positive association between melanin and symbiont density is most likely reflective of the dual roles of melanin synthesis in immunity and symbiont regulation. Melanin production can be used as a mechanism of symbiont shading in response to UV, reducing organismal stress (*Palmer, Bythell & Willis, 2011*). Further investigation of response of *A. poculata* to pathogens/immune threats, as opposed to measurement of constitutive immunity, will provide insight regarding the roles of melanin in immune responses, and clarify the mechanisms of symbiont-melanin associations.

Next, we considered associations between symbiont density, carbohydrate and lipid concentration and sperm density. Only carbohydrate concentration was positively associated with sperm density, suggesting that sperm release is at least in part resource limited. As gametogenesis in *A. poculata* begins earlier in the year (*Szmant-Froelich, Yevich & Pilson, 1980*), this association may be more indicative of the energetic costs of sperm release (spawning), rather than sperm production. Nutrient limitation has significant negative effects on multiple ejaculate traits (*Macartney et al., 2019*), reflective of the costs of sperm release. Further analysis of the association between energetic budget and sperm density throughout gametogenesis will provide improved understanding of temporal shifts in energetic allocation to reproduction.

In addition to sperm density, carbohydrate concentration was also positively associated with two metrics of immunity: melanin concentration and antibacterial activity. These associations are likely indicative of the high costs of both of these immune metrics. Melanin production is the result of a complex pathway; beginning with pathogen recognition and continuing through a series of protein cascades leading to the production of melanin (*Cerenius et al., 2010*; *Palmer, Mydlarz & Willis, 2008*). The complexity of the pathway is likely associated with high metabolic costs. To such end, previous studies have documented negative associations between lipid reserves and melanin concentration following immune stimulation (*Sheridan et al., 2014*). Similarly, antibacterial activity as measured here is the result of the action of many unique compounds. Corals and other cnidarians secrete a wide diversity of often complex antibacterial compounds (*Mitchell et al., 2019*; *Mydlarz et al., 2016*; *Palmer & Traylor-Knowles, 2012*), which may require significant energy to produce. In contrast, other metrics of immunity measured here, specifically antioxidants, are single molecules/compounds, and likely require significantly less metabolic input (*Traylor-Knowles & Connelly, 2017*). The low cost of investment in these components could explain the lack of strong correlation between our metrics of energetic budget and these immune parameters.

Finally, when considering the factors driving variation in immunity using both a multivariate and univariate approach, we found limited evidence for any association between sperm density and immunity. A moderate positive association between total phenoloxidase and sperm density was observed using correlational analyses. There is conflicting evidence regarding the link between reproduction and phenoloxidase in other invertebrates, with studies indicating both positive and negative associations (*Castella,*

*Christe & Chapuisat, 2009*; *Guo et al., 2021*). Further study of immune-reproduction associations in cnidarians will help clarify this relationship.

In total, our results were in contrast with our hypothesis; we observed no significant trade-offs between constitutive immunity and reproduction. We can posit several hypotheses to explain these observations. First, the timing of our experiment may not have properly reflected peak reproductive investment. Gametogenesis in *A. poculata* begins in early March-April and continues through June and July, with colonies spawning in early August/September (*Szmant-Froelich, Yevich & Pilson, 1980*). While we observed some association between sperm density and carbohydrate concentration indicative of resource allocation, is possible that this allocation may have been higher at earlier points of gametogenesis, as opposed to at the time of sampling in August (*Szmant-Froelich, Yevich & Pilson, 1980*). Consequently, the most pronounced trade-offs would have occurred earlier in the season, and tapered off by the point of sampling as resources availability increased and more resources could be allocated to immunity. Second, we chose here to measure sperm density due to sampling logistics. Sperm production is typically considered to be less energetically costly than production of eggs (*Hayward & Gillooly, 2011*; *Parker, 1970*; *Parker, 1982*). While we did observe some association between energetic resources and sperm, it is likely that these associations are more pronounced in females and consequently reproductive trade-offs would be more evident when considering females. Finally, our study does not account for multiple spawning events, specifically spawning events that may have occurred naturally prior to sample collection. Some coral species spawn multiple times throughout a season, and exact timing of *Astrangia* spawning in natural environments is unknown (*Szmant-Froelich, Yevich & Pilson, 1980*). Thus, it is possible that the collected *Astrangia* colonies had previously spawned *in situ*, and the induced events were a secondary release with reduced sperm density not reflective of true reproductive investment. Congruent with this possibility, several of our colonies had negligible measured sperm density. Future studies combining histology with immunological assays at different points of gametogenesis in both male and female colonies may clarify presence and timing of potential trade-offs.

It is also possible that nuances in our study design did not allow us to observe trade-offs. It must be noted that trade-offs occur in a multi-dimensional trait space filled with competed demands (*Lochmiller & Deerenberg, 2000*; *Stearns, 1989*), and our experimental analyses only captured two of these demands (*i.e.*, reproduction and immunity). It is highly likely that more complicated trade-offs, that involve other organismal demands (growth, maintenance, *etc.*,) may occur and were not captured by our study. Approaches that properly reflect the multi-dimensional trait space associated with resource allocation will be necessary to fully disentangle the relationship between immunity and other costly processes. Additionally, our study measured a limited number of immune metrics; it is possible that trade-offs exist between reproduction and immunity but involve other components of the immune system. The coral immune system is complex, involving many different components (receptors, signaling cascades, cellular and humoral effector responses, *etc.*), all of which have different relative costs (*Colditz, 2008*; *Ivanina et al., 2018*; *Lochmiller & Deerenberg, 2000*; *Seppala & Leicht, 2013*). It is therefore reasonable to

assume that trade-offs are not equivalent across components, as has previously been observed in other systems (*Adamo, Jensen & Younger, 2001*; *Albery et al., 2019*; *Gershman et al., 2010*; *Lochmiller & Deerenberg, 2000*). Future studies should incorporate more comprehensive metrics of immunity using methods such as gene expression or proteomics.

## CONCLUSIONS

In sum, our results fail to document notable associations between variation in symbiont density or energetic budget and metrics of host physiology (reproductive output and immunity). Furthermore, we find no evidence of trade-offs between reproductive output and immune activity. Still, our results are an important first step in broadening general understanding of resource allocation theory in cnidarians, which can be applied to other organisms facing rapid environmental changes. The information provided here provides an important preliminary framework for future studies of immunological trade-offs in marine invertebrates and the potential effects of variation in energetic budget on these patterns.

## ACKNOWLEDGEMENTS

The authors would like to thank Koty Sharp, Alicia Shickle, and members of the Roger Williams University Wet Lab for use of their facilities, assistance in collection of corals for this experiment, and execution of spawning/larval husbandry.

### Funding

This work was supported by startup funding from Texas State University to Lauren E Fuess. The funders had no role in study design, data collection and analysis, decision to publish, or preparation of the manuscript.

### Grant Disclosures

The following grant information was disclosed by the authors:
Texas State University.

### Competing Interests

The authors declare that they have no competing interests.

### Author Contributions

- Natalie Villafranca performed the experiments, analyzed the data, prepared figures and/or tables, authored or reviewed drafts of the article, and approved the final draft.
- Isabella Changsut performed the experiments, authored or reviewed drafts of the article, and approved the final draft.
- Sofia Diaz de Villegas performed the experiments, authored or reviewed drafts of the article, and approved the final draft.
- Haley Womack performed the experiments, authored or reviewed drafts of the article, and approved the final draft.

- Lauren E Fuess conceived and designed the experiments, performed the experiments, analyzed the data, prepared figures and/or tables, authored or reviewed drafts of the article, and approved the final draft.

## Field Study Permissions

The following information was supplied relating to field study approvals (*i.e.*, approving body and any reference numbers):

RI Department of Environmental Management, Division of Marine Fisheries

## Data Availability

The raw data and code used for all presented analyses are available in the Supplemental Files.

## Supplemental Information

Supplemental information for this article can be found online at http://dx.doi.org/10.7717/peerj.16586#supplemental-information.

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
