# Peer review of "Characterization of trade-offs between immunity and reproduction in the coral species Astrangia poculata"

_PeerJ, doi:10.7717/peerj.16586_

## Round 0.1 · original submission · Major Revisions

The manuscript has recieved a range of different review results, I urge the authors to follow all 4 reviewers' comments carefully and address the issues. One reviewer mentioned about missing description of immune assays and further questioned what are the immune/lipid/carb assays, and how are they used. The authors need to revise the manuscript precisely.

Reviewer 1 ·

Basic reporting

See additional comments

Experimental design

See additional comments

Validity of the findings

See additional comments

Additional comments

This manuscript describes an effort to quantify immunity and reproduction in a facultatively symbiotic stony coral. This work is part of an important effort to better understand trade-offs that may limit coral success in threatened environments, but the analyses performed do not thoroughly address the question. The authors picked several traits at a time to perform a series of trait-by-trait linear regressions. A more thorough and less biased approach would be a multivariate analysis regressing fitness (reproductive output) against all potential trade-offs or a correlation analysis of all traits simultaneously. The manuscript also lacks data transparency in terms of ranges of phenotypic values, especially within and among the two coral phenotypes (brown v white). Histograms and/or ordination analysis to summarize traits and allow clear comparisons among the two phenotypes would help and are described below. I ask the authors take a careful look at their statistical models to make sure the outputs agree with their conclusions (example below with a suspiciously negative value of an association described as positive). Lastly, I agree with the authors that this manuscript best serves the community as a launch pad for future experiments. Given that aim, I made suggestions throughout to improve methodological details so they can be repeated and to provide specific hypotheses in the discussion that can guide subsequent work.

General analytical suggestions:

(1) A summary of all parameters via some type of ordination analysis (NMDS, PCA etc) with samples colored by white vs. brown and loadings to indicate traits would be a great visual representation of all metrics and how they do or do not vary by host phenotype.
(2) Correlation matrix analysis among traits instead of pick-and-choose linear models with a correlation heatmap type figure demonstrating the relationships between all variables. The correlation matrix could also show histograms for each trait along one of the axes, which would address another data transparency issue.

Examples of ordination analyses and trait correlations in Muller et al 2021 (https://royalsocietypublishing.org/doi/10.1098/rspb.2021.0923) and Wright et al 2019 (https://doi.org/10.1111/gcb.14764).

Specific comments:

Line 28: Italicize genus species.
Line 47 vs. line 49 and throughout: Be consistent with tradeoff vs. trade-off as a noun.
Line 96: missing closing parenthesis
Line 115: What sizes were the colonies/how many polyps? Were entire colonies collected, or pieces of colonies? Provide that information here or later when describing the colonies/fragments used in the analyses.
Line 151: “slurry”
Line 142/154/throughout: Report centrifugation in g (i.e., convert according to the rotor radius of whatever centrifuge was used or provide rotor radius)
Line 149: The section is called “Protein Extraction”. Line 156 describes lipid and carbohydrate assays too from the “protein extraction”. Were lipids and carbs assayed from a different extraction or is the “protein extraction” a complete protocol for all physiological parameters? Please clarify.
Line 165 and throughout: Replace “u” with “µ” (if that is the intended symbol) throughout.
Line 246: Please cite and provide the version of R (the software) and provide the citation (not just version) for the IDE (RStudio), and double-check for any missed software packages used. All statistics were performed in base R?
Line 247: What was the basis of outlier detection?
Line 249: How is “energetic budget” defined?
Line 265: Do the asterisks represent interactions? Please clarify.
Line 272…: A figure would greatly help demonstrate these results. I’m especially curious to see what the association between symbiont density and lipid/carb looks like given the two types of colonies. Symbiont densities needed no transformation for normality? They would be extremely bimodal because of inclusion of white and brown colonies in the same analysis, yes? More data transparency showing distributions and regression analyses are needed to fully appreciate this point. In general I’m cautious against including symbiont density only as a quantitative metric when it also exists as a categorical variable (white v. brown), especially without access to the knowledge of distributions of symbiont densities within and between the two categories of corals. I can imagine a scenario where carbohydrate content (or lipid or whatever other metric) is significantly associated with symbiont density in the brown colonies but not in the white colonies (where it’s more associated with some other metric, like predation that wasn’t captured by this study). A hypothesis like this is relevant to this study, but untested as far as I can tell without knowing the distributions of symbiont densities.

Line 273 and elsewhere: What are the “est” values? Coefficients?

Line 282…: “Melanin concentration was significantly positively associated with both lipid and carbohydrate concentration…” and, yes, the slope looks positive in Figure 1, but the “estimate” values (here I assume these are coefficients of the linear model and may need clarification as indicated above) are negative. Table 2 melanin estimate = -8.1e-3 and Table 3 melanin estimate = -4.84e-3. If these negative estimate values represent positive associations, please clarify.

Line 325…: Another consideration to discuss is that these corals are not (to your knowledge) under an current immune threat. Melanization may play another function besides immunity whereas the other immune assays represent specialized defenses. Or melanin may be primarily involved in immunity and the only component of that immunity that’s “front loaded” in these corals whereas the other metrics are more responsive. Given the priority of this manuscript for furthering future research, this point is important guidance for future studies to potentially include an immune challenge to see how these patterns persist under a real threat.

Line 309: Can you offer any guidance to future experiments who may wish to test the hypothesis about symbiont densities compensated by heterotrophy? Have other studies indicated that white colonies engage in heterotrophy more than brown?

Line 312: Can you offer any suggests on what other metrics would better capture host energetic budget? I see “metabolomics” and “carbon transfer” listed below as general methods, but I’m wondering if you have a specific hypothesis about what your assays may have missed (e.g., a type of carb or lipid not well quantified by your assay or some other essential nutrient like iron).

Line 331–336: I’m not following this explanation. If reproductive investment in spring is positively associated with sperm production in late summer, there should be no conflict in the timing of your experiment with respect to getting a metric of overall “reproductive output.” If reproductive investment in spring is unrelated (or somehow even negatively correlated) to sperm production in late summer, the spring investment phase would be unrelated to fecundity/fitness and therefore not a good metric for assessing trade-offs. So yes, the energetic investment phase could have been higher earlier in the season, but if that higher investment isn’t reflected in sperm production, what is the point from the coral’s perspective or yours as a predictor of reproductive potential?

Line 352: Clarify what is meant by “our experimental analyses only captured two of these traits.” The study design measures many traits as far as I can tell (sperm, sym density, multiple immune metrics, protein, carb, lipid…)

Line 358–359: “… different processes with different relative costs”… it would be helpful for future studies to provide an example

Figure 1 legend: “classification” typo
Figure 1b: Are the y-axis labels correct? It looks like the bottom is 4e5 (missing negative sign?)
Table 5: Are these predictors supposed to say “carbohydrate” instead of “lipid”?

Reviewer 2 ·

Basic reporting

Overall, I think that this is a very well written and clear study. I really applaud the authors for publishing negative results and putting them into context, with positive results. The writing was clear and professional English was used throughout. There was only one area where I found the language to stray a little and that is in Lines 245-268-"statistical Analysis" section. In particular I recommend the following:

Editing lines 248-249: changing it from "First we" to " First the effects of ................were examined using linear models."




For the discussion:

Overall I think that the discussion could be flushed out more to put things into context more. In particular thinking about the environment that these coral come from. Are these areas nutrient rich? Light poor or high? Since most of the context was set based on tropical corals, which would be a very different environment, I think adding in more context about the astrangia environment would be helpful.

- I suggest sub-headers for each part of the results that you are reporting on.
- I was struck that most of the references were based on tropical corals, and I wonder if there are more astrangia references that could be used to put things into better context. I am not up-to-date on this area of literature.
-What about the environment? This was never discussed in the discussion and I think that this could be an important component to the trade-off idea and also why so little differences were seen.
-Also, what about other immune markers? Could there be ones that more specific to astrangia that may be more important?
-Since much of the hypotheses is taken from tropical corals (which I completely understand why the authors did this) there may be many areas of missed knowledge by not examining or at least acknowledging the unique environment that they live in and that this would have to have major influence on their symbioses and immunity.

Experimental design

Overall the scope of the research was well laid out and the questions were meaningful and insightful. I appreciated the detailed methods that were presented. One qquestion below:


Line 252-253: how was average sperm output per polyp determined? Was this the same as sperm density mentioned above? If yes, than i would just say Sperm density, if no, then I would add part to define this so that the reader can better understand it.

Validity of the findings

As I said before I appreciate that the authors are reporting on negative results, along side the positive ones. However I think a deeper dive into the context is warranted, in particular exploring in the discussion possible environmental effects.

I also think that adding in a summary figure at the discussion would be helpful, especially since many different markers were used and because this is a really a hypothesis/ideas paper. I think that it would help the reader to understand why this is interesting and novel. Or maybe since the authors rely on the tropical coral literature, do a summary figure that compares astrangia to tropical models for these trade offs? I think that it could add more context to this study and would help the reader to really understand the uniqueness of these findings.

·

Basic reporting

This paper is well written and presents its aims well. Relevant literature is reviewed and referenced. The paper is well structured and presents its limitations effectively. Tables and figures are provided, as well as the raw data and code used for the statistical analysis. The results are presented in conjunction with the authors’ questions. The framing of this paper is really unique and I enjoyed reading it!

Experimental design

The experiment presented here was novel and relevant to current resource allocation research. Methods were presented in great detail and were well explained in the text. Replication was sufficient for the corresponding statistical analyses. In either the introduction or the methods, it would help to outline why the authors chose to study their specific immune variables (total phenoloxidase activity, melanin, catalase, peroxidase, and antibacterial activity). I am not familiar with the importance of these variables, so some clarification or justification of why these were chosen for this experiment would be helpful.

Validity of the findings

I commend the authors for their clear and thorough presentation of their findings. It is often difficult to present results that are not significant, but the authors did a great job discussing why they believe some results were not significant, as well as caveats to their experimental design. The authors thoroughly addressed the nuances in their study design and explicitly stated that it is possible that tradeoffs might be observed under different circumstances, by measuring different physiological variables or by evaluating female tradeoff potential.

Additional comments

- Tables 1, 4, 5, and 6 (any table that does not include a significant p-value) can be moved to the supplementary material.
- Line 52 “Among the most well documented…” - this sentence would be better supported by adding an example of well documented tradeoffs between immunity and reproduction.
- Throughout the methods section, change uL to μL
- Line 179 - change forty microliters to 40 μL.
- Same comment above to lines 192, 218, 222, 223, 227, 237.
- Great job including the stats information in the results section!
- Lines 317-18 “The observed association between melanin concentration and total lipids and carbohydrates is likely indicative of the high cost of melanin production” - Based on Figure 1, it looks like with higher melanin concentrations, there are higher lipid or carbohydrate concentrations. If melanin is so costly to produce, wouldn’t there be less carbs and lipids, as the corals would be allocating their energy towards making melanin?
- It would be interesting to compare the results presented here with those found in Harman et al. (2022). Harman et al. examined immunity metrics from Astrangia in RI and discussed resource allocation in symbiotic and aposymbiotic corals. This paper would be relevant to reference in the discussion section. See paper here: https://doi.org/10.3354/dao03695

·

Basic reporting

The manuscript is very well written, and I generally have few grammatical comments/suggestions below. One general comment about the introduction was that the first paragraph was largely dedicated to unspecified taxa, and the coral-specific background was relatively short. One aspect that was particularly missing was a description of the immune assays that were used in the study – what are the immune/lipid/carb assays, and how are they used as a proxy for immune function? The introduction mentions several coral transcriptomic studies that examined immune activity, so why not use transcriptomics for this study? Some justification will go a long way, and will help readers to understand the reasoning behind using these particular assays.

I also noticed that units were either abbreviated or spelled out entirely throughout the manuscript. I would suggest following the journal guidelines, which I’m guessing would be abbreviated.

Finally, it would be nice to see additional figures in the paper. Perhaps a photo of your corals spawning, especially since many readers would not be familiar with Astrangia? I know that the majority of statistical tests were not significant, but a panel of non-significant relationships that were tested would provide some additional context to Figure 1. At present, it feels a little reductive, and underrepresents the amount of work that went into this study. You could also consider combining the tables into one large table that distinguishes between different tests (symbiont density, lipids, carbs, lipids*sym state, carbs*sym state, sperm). This may also happen naturally if you run multivariate tests.

Experimental design

Once again, the writing in the methods prior to the statistical design are well-written and thoroughly explained. One thing I am confused about is the choice of running one-way tests for each metric? Why not run a multivariate test? Do you expect that the various response variables are highly collinear? And if so, should you be testing for that and reporting those results? The model described on L264-265 seems the most appropriate, though it is unclear whether all immune assays were run as a multivariate test vs multiple univariate tests. I would suggest reminding the readers what ‘immune activity’ includes. Some clarity would be helpful on why the particular statistical tests were chosen.

Second, are all of these relationships assumed to be linear? Your use of linear models would suggest so, but what if the relationships are non-linear? As a result, the statistical design comes off as clunky.

Validity of the findings

The authors found few significant relationships between their measured metrics and immune function/reproductive output besides melanin vs energetic budget. Their interpretation focused on a few potential explanations: choice of metrics measured, energy expenditures in molecule synthesis, and mis-timing of gametogenesis vs sampling. The first and last points unfortunately have the most profound implications for the study. The authors state that gametogenesis occurs potentially months before their samples were collected (or that spawning already occurred), and that females/eggs are more likely to demonstrate energetic differences. I agree with this statement, and without quantitative assessments of egg production/quality, it becomes difficult to interpret their findings beyond experimental limitations.

Taken together, I unfortunately think this study has fundamental flaws, and additional experiments/data collection should be considered to make this study appropriate for publication. In particular, fecundity assessments and more-comprehensive approaches like transcriptomics would be valuable. Second, if corals could be held ex situ from prior to gametogenesis through spawning, you could make additional observations and rule out mis-timing of sampling with spawning activity. For these reasons, I recommend that this manuscript be rejected as it stands.

Additional comments

L28: Species name should be italicized
L115: How many colonies of each morphotype? And do you know if they are distinct genotypes?
L131: How big were these colonies? Given that you counted the number of polyps on each, I’m assuming they are small? Would it make more sense to use fragments instead, or are the colonies that small in the wild?
L151: slurry
L152: The journal may have standard abbreviations to use for units here and throughout
L169: [pH 7.8]
L180: What is a ½ well 96-well plate? Does that refer to a 48-well plate, or were the individual wells a smaller volume?
L189: Check journal formatting guidelines for parentheticals within parentheticals. I’m guessing it should be something like: (POX; ref1; ref2)
L228: Please tell me this assay smells like vanilla…I feel like I need to know
L239: is there a difference between the uncovered hot water bath in this step vs the hot water bath described in L224?
L247: Were the assay outliers from the same sample?
L260: The way this sentence is written, it looks like the two models are combined somehow. I would add a comma before and
L295: Remove the second comma

---

## Round 0.2 · Minor Revisions

Reviewers suggested for the minor revision, I advise authors to follow their comments carefully and revise the draft. It will be better if you could clear discussion on the word "trade-offs between immunity and reproduction". In addition, please add the scientific position for the given scientific name in the title, Phlyum,or Order or Class, etc.

Reviewer 1 ·

Basic reporting

I appreciate the authors’ improvements in terms of data transparency, methodological details, statistical analysis, and clarity of discussion points. I have only minor points described below.

Minor comments:
Line 171: “Vibrio”

Throughout the manuscript, clarify phrasing about the hypotheses and conclusions of this study with respect to what “causes” or “affects” what. To my understanding, the study investigates associations between variables without a clear direction. For example, symbiont density may be associated with an immune metric but one could hypothesize that the higher symbiont density “impacts/causes” the higher immunity or vice versa. For example:
Line 299: “symbiont density had limited impacts on host immunity…” is that within the scope of this study? I think the next line “not detecting any differences between white/brown” is accurate.
Line 311: “the effects of symbiont density, carbohydrate and lipid concentration on sperm density…” again – is it within the scope to describe sperm density as an effect of sym/carb or is the study testing an association?
Line 386: “fail to document notable effects of variation in symbiont density and energetic budget on coral physiology…” Again, I’m not sure the ‘effects of variation on symbiont density’ were being directly measured in this study and the language of “coral physiology” is vague here. What exactly about the physiology was tested for association with sym density?

Line 388: Clarify. “… no evidence of immune-associated trade-off.” Trade-off with what? Reproductive output?

Experimental design

no comment

Validity of the findings

no comment

Additional comments

no comment

---

## Round 0.3 · accepted · Accept

The authors have followed all reviewers' comments and revised well. This manuscript can be accepted and considered for publication.

Reviewer 1 ·

Basic reporting

No need to re-review.

Experimental design

No need to re-review.

Validity of the findings

No need to re-review.

Additional comments

No need to re-review.